# IMPROVING FEATURE ALIGNMENT IN CONVNETS USING CONTRASTIVECAMS AND CORE-FOCUSED CROSS-ENTROPY

## ABSTRACT

Despite the ubiquity of modern deep learning, accurate explanations of network predictions remain largely elusive. HiResCAM is a popular interpretability technique used to visualize attention maps (i.e., regions-of-interest) over input images. In this paper, we theoretically show a limitation of HiResCAM: the HiResCAMs for a given input are not uniquely determined, allowing an arbitrary spurious shift by a common matrix $M$ while corresponding to the same prediction. We further propose *ContrastiveCAMs*, which are invariant to the spurious shift $M$ hence improving robustness of explanations, while additionally providing granular class-versus-class explanations. With the additional granular explanations, experiments reveal that networks often focus on regions unrelated to the class label. To address this issue, we leverage the knowledge of core image regions and propose *Core-Focused Cross-Entropy*, an extension of cross entropy, which encourages attention on core regions while suppressing unrelated regions, improving feature alignment. Experiments on Hard-ImageNet and Oxford-IIIT Pets show that ContrastiveCAM provides more faithful attention maps and our method effectively improves feature alignment by primarily extracting predictive performance from core image regions.

## 1 INTRODUCTION

The vast applications of convolutional neural networks in safety-critical domains such as medical imaging (Kc et al., 2021; Rajpurkar et al., 2017), forensic investigation (Murthy and Siddesh, 2023) and self-driving (Kim and Canny, 2017) make accurate (a.k.a faithful) interpretations of their predictions paramount (Haufe et al., 2024). Approaches to explain predictions include feature-attribution based interpretability techniques (Zhou et al., 2016; Selvaraju et al., 2017; Draelos and Carin, 2020), input-based interpretability with saliency maps (Simonyan et al., 2013; Smilkov et al., 2017), and more recently, mechanistic interpretability for image circuit discovery (Olah et al., 2020).

In addition to faithful interpretability, ensuring that only target-relevant (a.k.a. core) regions influence model predictions is a critical determination to make. A model-agnostic approach for evaluating the impact of core regions involves input ablation experiments as introduced in recent work on Core Risk Minimization (Singla et al., 2022; Moayeri et al., 2022). Images are modified to systematically corrupt core regions, following which the change in performance is reported. Singla and Feizi (2022) demonstrate that both convolutional and transformer-based architectures are vulnerable to learning non-core regions of the input, caused by features like co-occurring backgrounds. These encourage learning 'tricks' – shortcuts to learning that improve in-distribution accuracy while inhibiting generalization over core features (Geirhos et al., 2020). A concrete example of shortcut learning is illustrated within the introduction of Invariant Risk Minimization (Arjovsky et al., 2020).

In this work, we develop and leverage faithful interpretability to encourage feature alignment in convolutional models. We theoretically observe that HiResCAMs (Draelos and Carin, 2020) may not explain true factors that contribute towards predictions as a consequence of *softmax* activation. Specifically, we prove that HiResCAMs are not uniquely determined and admit arbitrary, spurious shifts by a common matrix $M$ while corresponding to the same prediction (Theorem 3.2). This spurious shift from $M$ can, in principle, completely corrupt HiResCAM explanations. To remove this redundancy, we propose *ContrastiveCAMs* (Definitions 3.3, 3.4), resulting in attention maps that

are invariant to the aforementioned spurious shift while additionally providing granular class-versus-class explanations. Using class-versus-class comparisons, we experimentally reveal circumstances wherein different comparisons leverage different regions to base their predictions. Further, these differing regions do not always correspond to core regions of the input image, i.e., there are spurious contributions. We demonstrate that cross entropy loss encourages leveraging these unrelated regions, especially in settings where the target represents a small portion of the image (Section 4.1). Finally, we propose a modification to cross-entropy, termed *Core-Focused Cross-Entropy* (Definition 4.5), which: a) suppresses user-specified non-core regions despite the presence of spurious factors, and b) generates contrast within user-specified target regions to solve for the underlying classification task. This improves feature alignment by encouraging the model to learn target-relevant features only.

We demonstrate the effectiveness of our proposed method by reporting experimental results in multiclass, multiple-class, and binary classification settings. We supplement this evidence by showing that core-focused models may be trained competitively even with coarse or auto-generated masks, and that they outperform backbones trained using cross-entropy in downstream segmentation tasks.

## 1.1 RELATED WORK

**Feature Attribution in Convolutional Networks.** A prominent family of interpretability techniques stems from the seminal CAMs (short for Class Activation Mappings) (Zhou et al., 2016) literature. CAMs help identify regions-of-interest in the form of attention maps. It's success led to the introduction of a vast set of derivative works, that extend CAMs in various ways (Selvaraju et al., 2017; Chattopadhay et al., 2018; Wang et al., 2020; Draelos and Carin, 2020).

**Representation Learning.** Arjovsky et al. (2020) introduces the notion of predictors that learn feature representations that are invariant to spurious factors. Bau et al. (2017) quantifies the interpretability of learned representations in convolutional models by evaluating hidden units within convolutional layers on segmentation tasks. Recently, Zou et al. (2023) motivates neuroscience-inspired top-down approached for inducing interpretability. It encourages the analysis of representations (representation reading) and it's subsequent modification (representation control).

**Feature Alignment.** Spurious factors in images encourage extracting predictions from unrelated regions, termed *shortcuts*, and are discussed extensively by Geirhos et al. (2020). Feature alignment seeks to ensure predictions are made using relevant features only, and is deeply connected with robustness in neural networks (Wang, 2023). Preventing shortcut learning is thus a crucial goal of feature alignment. Approaches to alignment include region masking (Kc et al., 2021), tiered training (Aniraj et al., 2023), and regularization via saliency maps (Ismail et al., 2021), each having an empirical focus. For a thorough exposition to recent advancements and challenges in interpretability-guided feature alignment, we direct the reader to Weber et al. (2023) and Gao et al. (2024).

## 2 PRELIMINARIES

**Notation.** We denote vectors using bold lowercase letters (e.g., $\mathbf{v}$), matrices using uppercase letters (e.g., $M$), and tensors using bold uppercase letters (e.g., $\mathbf{T}$), with partial indexing implying selection of the subtensor across the remaining subsequent dimensions (e.g., $\mathbf{T}_i \in \mathbb{R}^{b \times c}$ for $\mathbf{T} \in \mathbb{R}^{a \times b \times c}$). We use the operator $\odot$ to represent elementwise multiplication, and define $[C] := \{1, 2, \ldots, C\}$.

**Setup.** In this paper, we consider image classification tasks. The dataset $\mathcal{D} = \{(\mathbf{X}^{(i)}, \mathbf{y}^{(i)})\}_{i=1}^n$ contains image-label pairs where images are represented using rank-3 tensors $\mathbf{X}$ consisting of two spatial dimensions and one channel dimension, and labels are one-hot vectors $\mathbf{y} \in \mathbb{R}^C$, where $C$ denotes the total number of classes in the dataset. A neural network $f$ is trained to learn the relation between the images $\mathbf{X}$ and labels $\mathbf{y}$. The output of $f$ contains $C$ *logits*: $f_c, c \in [C]$. Let $\sigma(\cdot)$ be the softmax function, $\tilde{f}(\mathbf{X}) = \sigma(f(\mathbf{X}))$ is then interpreted as the class-specific *probability predictions*. The standard training procedure is to optimize a cross-entropy loss function so that $\tilde{f}(\mathbf{X})$ matches the label $\mathbf{y}$ as closely as possible for each training image-label pair.

In prominent approaches such as VGG (Simonyan and Zisserman, 2015), ResNet (He et al., 2016) & ViT (Dosovitskiy et al., 2020), the neural network $f$ mainly consists of two consecutive parts, a

backbone module $g$ followed by a classifier $h$: $f = h \circ g$. In this paper, we focus on convolutional neural networks, i.e., the backbone $g$ is convolutional. We denote the output of the backbone $g$ as $\mathbf{A} \in \mathbb{R}^{d_0 \times d_1 \times d_2}$, termed as feature embedding (a.k.a. feature maps) of the image, where $d_0$ is the number of features (a.k.a. channels) and $d_1$ and $d_2$ are the spatial dimensions of the final convolutional layer. The feature embedding $\mathbf{A}$ is then reduced to a vector $\mathbf{z}$, either by flattening $\mathbf{z} = vec(\mathbf{A})$, or by Global Average Pooling (GAP). $\mathbf{z}$ is then processed by the classifier $h$, which outputs the logits $f$, that are passed through *softmax* to obtain the class prediction vector, denoted $\tilde{f}$.

The recent trend is that the classifier $h$ becomes as simple as a single layer, such as in ConvNext (Liu et al., 2022), ViT (Dosovitskiy et al., 2020), EfficientNet (Tan and Le, 2019), ResNet (He et al., 2016) & DenseNet (Iandola et al., 2014):

$$h(\mathbf{z}) = W\mathbf{z} + \mathbf{b}, \tag{1}$$

This simplification of $h$ is largely due to the fact that the backbone $g$, which encapsulates the bulk of the model's predictive power, extracts high quality and comprehensive features $\mathbf{A}$, based on which a single layer is enough to obtain accurate final predictions. In this paper, we assume that the classifier is of the form in Eq. (1).

**HiResCAMs.** HiResCAMs (short for High-Resolution Class Activation Maps), introduced in (Draelos and Carin, 2020), is a method designed to provide interpretable explanations of convolutional neural networks. It renders the contribution of each spatial location in an image to the final logit output $f_c$, thereby revealing which regions are most critical to the models prediction. Specifically, given a feature embedding $\mathbf{A}$ of an image $\mathbf{X}$ and a class index $c \in [C]$, the HiResCAM is defined as:

$$\mathbf{CAM}_c^{\text{HiRes}} = \sum_{j=1}^{d_0} (\nabla_{\mathbf{A}_j} f_c) \odot \mathbf{A}_j, \qquad \mathbf{CAM}_c^{\text{HiRes}} \in \mathbb{R}^{d_1 \times d_2} \tag{2}$$

$\mathbf{CAM}_c^{\text{HiRes}}$ shares spatial dimensions with the backbone output $\mathbf{A}$. Each element within $\mathbf{CAM}_c^{\text{HiRes}}$ represents a contribution to the logit output $f_c$ from a corresponding patch within the original image. A higher absolute value implies a greater contribution.

HiResCAMs have been widely used for incorporating explainability in a variety of tasks, such as CT scan abnormality classification (Draelos and Carin, 2022), malware visualization (Brosolo et al., 2025), coffee leaf rust classification (Chavarro et al., 2024), counterfeit banknote detection (Pachón et al., 2023) & flow estimation (Chen and Wu, 2025).

Particularly, for single-layer classifiers $h$, Draelos and Carin (2020) show that the expression of HiResCAMs, Eq. (2), can be simplified and has the following close connection with output logits $f_c$:

$$f_c(\mathbf{X}) = \sum_{i=1, j=1}^{d_1, d_2} \mathbf{CAM}_{c,i,j}^{\text{HiRes}}(\mathbf{X}) + \mathbf{b}_c, \qquad c \in [C]. \tag{3}$$

Each logit $f_c$ is the summation of the HiResCAM over its spatial dimensions, up to a scalar $\mathbf{b}_c$.

## 3 Contrastive Class Activation Maps

In this section, we first discuss the theoretical limitations of HiResCAM in explaining model predictions, and then introduce a surrogate method, ContrastiveCAM, which offers more faithful and class-specific explanations.

**HiResCAMs Admit Spurious Shifts.** A key observation is that HiResCAMs are only related to *logits* $f$, not *probability predictions* $\tilde{f} = \sigma(f)$ belonging to each class, see Eq. (3). The drawback is that, for the same probability prediction $\tilde{f}$, there are infinitely many possible logit outputs $f$, hence infinitely many HiResCAMs, each of which explain the same prediction differently. This drawback arises intrinsically from the nature of the *softmax* function.

**Proposition 3.1** (Contrastiveness of *softmax*). *The softmax function is invariant to a universal shift of all its input components:*

$$\sigma(\mathbf{x}) = \sigma(\mathbf{x} + a\mathbf{1}_C) \qquad \forall \mathbf{x} \in \mathbb{R}^C, \ a \in \mathbb{R} \tag{4}$$

*Proof.* All proofs are deferred to Appendix A. □

This invariance to $a \in \mathbb{R}$ is amplified to a matrix $M \in \mathbb{R}^{d_1 \times d_2}$ when assessing HiResCAMs.

**Theorem 3.2.** *HiResCAM explanations* $\mathbf{CAM}^{\mathrm{HiRes}} \in \mathbb{R}^{C \times d_1 \times d_2}$ *corresponding to probability predictions* $\tilde{f}(\mathbf{X}) \in \mathbb{R}^C$ *are not uniquely determined, admitting a universal shift of class-level explanations* $\mathbf{CAM}_c^{\mathrm{HiRes}}$ *by an arbitrary matrix* $M \in \mathbb{R}^{d_1 \times d_2} \ \forall c \in [C]$.

$$\tilde{f}(\mathbf{X}) = \sigma \left( \sum_{i=1,j=1}^{d_1,d_2} \mathbf{CAM}_{:,i,j}^{\mathrm{HiRes}} + \mathbf{b} \right) = \sigma \left( \sum_{i=1,j=1}^{d_1,d_2} \overline{\mathbf{CAM}}_{:,i,j}^{\mathrm{HiRes}} + \mathbf{b} \right) \quad \forall M \in \mathbb{R}^{d_1 \times d_2} \quad (5)$$

*Where* $\overline{\mathbf{CAM}}^{\mathrm{HiRes}}$ *is defined as:*

$$\overline{\mathbf{CAM}}_c^{\mathrm{HiRes}} := \mathbf{CAM}_c^{\mathrm{HiRes}} + M \qquad \forall c \in [C] \qquad (6)$$

Thus explanations from HiResCAMs are accurate only upto a summand $M$ which is unknown. These explanations may be misleading, and *fail to guarantee a faithful interpretation* of the model prediction. An example of such a misinterpretation is illustrated in Figure 1.

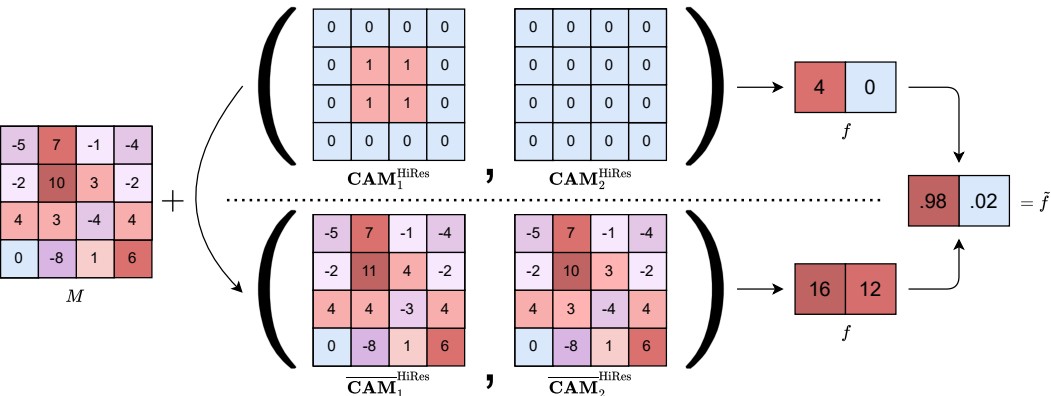

Figure 1: Shifting $\mathbf{CAM}^{\mathrm{HiRes}}$ by arbitrary matrix $M$ results in a change to explanations $\overline{\mathbf{CAM}}^{\mathrm{HiRes}}$ which subsequently changes the corresponding logit vector. However, the model's final prediction probabilities are identical and remain unchanged.

To remove this redundancy, we define a contrastive representation of HiResCAMs, which recovers faithful attention maps at the class probability level.

**Definition 3.3** (ContrastiveCAMs). Given a set of classes $[C]$ with $c_t$ being the index of the target class for a given image, ContrastiveCAM is defined as follows:

$$\mathbf{CAM}_{c_t}^{\mathrm{Cntrst}} := \left\{ \mathbf{CAM}_{(c_t,c')}^{\mathrm{Cntrst}} : c' \in [C] \setminus c \right\}, \quad \mathbf{CAM}_{(c_t,c')}^{\mathrm{Cntrst}} := \mathbf{CAM}_{c_t}^{\mathrm{HiRes}} - \mathbf{CAM}_{c'}^{\mathrm{HiRes}} \quad (7)$$

Further, we also reconstruct single-class interpretations of ContrastiveCAMs:

**Definition 3.4** (Class-Reconstructed ContrastiveCAMs). Given a set of classes $[C]$ with $c_t$ being the index of the target class for a given image, reconstructed ContrastiveCAMs are defined as follows:

$$\mathbf{CAM}_{c_t}^{\mathrm{Recon}} := \frac{1}{C} \sum_{c=1}^{C} \mathbf{CAM}_{(c_t,c)}^{\mathrm{Cntrst}} = \mathbf{CAM}_{c_t}^{\mathrm{HiRes}} - \frac{1}{C} \sum_{c=1}^{C} \mathbf{CAM}_c^{\mathrm{HiRes}} \qquad (8)$$

$\mathbf{CAM}_{c_t}^{\mathrm{Recon}}$ thus removes redundancy $R = -1/C \cdot \sum_{c=1}^{C} \mathbf{CAM}_c^{\mathrm{HiRes}}$. We report the ratio of redundancy to the original explanation as $\gamma = \|R\|_F / \|\mathbf{CAM}_{c_t}^{\mathrm{HiRes}}\|_F$ for various datasets in Table 1.

Crucially, ContrastiveCAMs are invariant to spurious contributions as exposed by Theorem 3.2.

**Theorem 3.5** (ContrastiveCAMs are $M$-invariant). *Let* $\mathbf{CAM}^{\text{HiRes}}$ *and* $\overline{\mathbf{CAM}}^{\text{HiRes}}$ *be two HiResCAMs corresponding to probability predictions* $\tilde{f}(\mathbf{X}) \in \mathbb{R}^C$ *such that:*

$$\overline{\mathbf{CAM}}_c^{\text{HiRes}} = \mathbf{CAM}_c^{\text{HiRes}} + M \qquad \forall c \in [C] \tag{9}$$

*Then, for every* $M \in \mathbb{R}^{d_1 \times d_2}$, *it holds that:*

$$\mathbf{CAM}^{\text{Cntrst}} = \overline{\mathbf{CAM}}^{\text{Cntrst}} \quad \text{and} \quad \mathbf{CAM}^{\text{Recon}} = \overline{\mathbf{CAM}}^{\text{Recon}} \tag{10}$$

**Class-versus-Class Explanations.** While explanations from the CAM-family only involve visualizing $f_{c_t}$, softmax activation uses every logit in computing class probabilities. Making inferences based on individual logits may thus misinterpret the internal model state, as the training objective induced by cross-entropy loss over softmax activation is to maximize the **difference between class logits**, see Eq. (44). We demonstrate the value of additional granularity provided by pairwise explanations by reporting observations on a three-class subset of Hard-ImageNet in Figure 2.

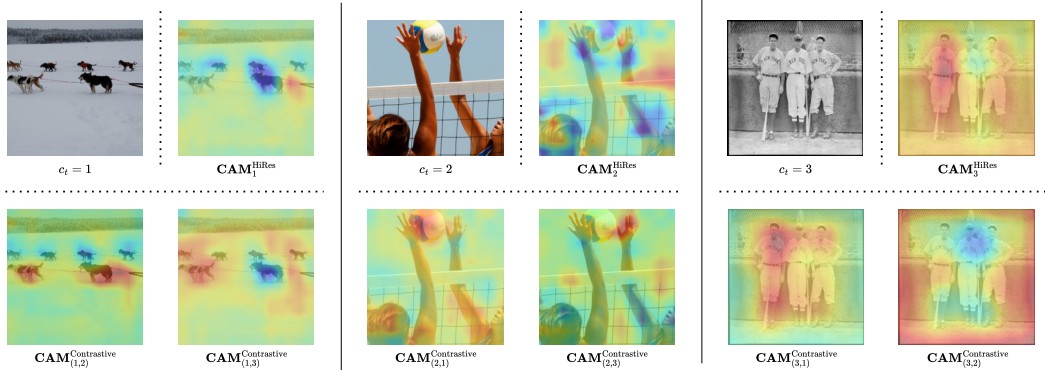

Figure 2: We plot ContrastiveCAM and HiResCAM explanations on a ResNet-18 model trained to classify: ('dog sled', 'volleyball', 'baseball player'), ordered by label index. ContrastiveCAMs reveal circumstances wherein: a) regions that contribute towards prediction are hidden by HiResCAMs, and b) differing parts of the image contribute towards various class-versus-class predictions.

From Figure 2, we also observe that the model often leverages irrelevant regions (e.g., environmental cues), to inform predictions. Following Moayeri et al. (2022), we refer to these regions as *non-core regions*. In principle, *core regions* are those that causally influence the prediction (modification of this region could mean the ground truth itself may change), while *non-core regions* represent spurious correlations – modifications to these regions do not change the ground truth labels.

Table 1: ContrastiveCAM explanations bifurcated by core-region maps across various datasets. The average contributions of core / non-core regions and ratio of redundancy removed is reported below.

| **Dataset** | Core ($\uparrow$) | Non-Core ($\downarrow$) | $^{\text{Core}}/_{\text{Total}}$ ($\uparrow$) | Redundancy ($\gamma$) | Accuracy (%) |
|---|---|---|---|---|---|
| Hard-ImageNet | 14.817 | **42.138** | .2601 | .201 | 95.73 |
| Oxford-IIIT Pets | **3.925** | 2.150 | .6461 | .367 | 99.34 |
| PASCAL VOC | 1.581 | **1.719** | .4791 | $-$[1] | 87.32 |

This undesired influence is consistently observed, as evidenced by high overall non-core contribution in Table 1 above. Despite strong accuracy, large contributions arise from non-core regions.

## 4 LEARNING WITH CONTRASTIVECAMS

The dependency on non-core regions observed above is evidence of misalignment, which inhibits generalization. In this section, we prove a desirable theoretical property of ContrastiveCAMs and leverage it to incorporate interpretability within model optimization, mitigating this weakness.

Specifically, we prove that any *input-dependent* change to probability predictions $\tilde{f}$ (e.g., caused by updating model weights) is precisely reflected by a proportionate change to $\mathbf{CAM}^{\mathrm{Cntrst}}_{c_t}$.

**Proposition 4.1** (Correctness of ContrastiveCAMs). *Softmax-activated class probabilities $\tilde{f}$ can be expressed as a direct function of ContrastiveCAMs and the bias vector.*

$$\tilde{f}_{c_t}(\mathbf{X}) = \left( \sum_{c=1}^{C} \exp \left( \mathbf{b}_c - \mathbf{b}_{c_t} - \sum \mathbf{CAM}^{\mathrm{Cntrst}}_{(c_t,c)} \right) \right)^{-1} \qquad \forall c_t \in [C] \qquad (11)$$

*Where* $\mathbf{CAM}^{\mathrm{Cntrst}}_{(c_t,c_t)} = \mathbf{0}_{d_1 \times d_2}$.

By zero-ing the final bias vector (i.e., $\mathbf{b} := \mathbf{0}_C$ for $h$ *only*), we can precisely disassociate the role of specific regions in computing cross-entropy. We leverage this property to study feature misalignment, and later in our proposed modification of cross-entropy to penalize the use of non-core regions.

## 4.1 CROSS-ENTROPY CAN MOTIVATE FEATURE MISALIGNMENT

To encode core-region information, for each sample from our dataset of size $N$, we extend dataset $\mathcal{D}$ by specifying a binary mask $H$, which indicates whether or not downsampled regions from the input image may be used to determine the prediction.

$$\mathcal{D} := \{(\mathbf{X}^{(i)}, (H^{(i)}, \mathbf{y}^{(i)}))\}_{i=1}^{N} \quad \text{where} \quad H_{jk} := \begin{cases} 1 & \text{region contains target} \\ 0 & \text{region doesn't contain target} \end{cases} \forall j, k \in [d_1], [d_2]$$

We can restate cross-entropy as a function of ContrastiveCAMs and core-region information in $\mathcal{D}$.

**Proposition 4.2.** *Given bias-free classifier $h$, we can precisely associate the impact of specific regions, encoded by binary mask $H$, to the computation of cross-entropy loss.*

$$\mathcal{L}_{\mathrm{CE}}(f(\mathbf{X}), \mathbf{y}, H) = \log \left( \sum_{c=1}^{C} \exp \left( -\sum H \odot \mathbf{CAM}^{\mathrm{Cntrst}}_{(c_t,c)} - \sum (1-H) \odot \mathbf{CAM}^{\mathrm{Cntrst}}_{(c_t,c)} \right) \right) \tag{12}$$

**Remark 4.3.** *Equivalently, we disassociate the logit and use the standard cross-entropy formulation:*

$$\mathcal{L}_{\mathrm{CE}}(f(\mathbf{X}), \mathbf{y}, H) = \mathcal{L}_{\mathrm{CE}} \left( \sigma \left( -\sum_{i=1,j=1}^{d_1,d_2} \underbrace{H \odot \mathbf{CAM}^{\mathrm{Cntrst}}_{(c_t,:),i,j}}_{\mathrm{core}} + \underbrace{(1-H) \odot \mathbf{CAM}^{\mathrm{Cntrst}}_{(c_t,:),i,j}}_{\mathrm{non-core}} \right), \mathbf{y} \right) \tag{13}$$

We observe from Proposition 4.2 that cross-entropy loss does not inherently favor using the core or non-core regions for classification. Provided the prediction is accurate with high confidence, error remains low. This presents a theoretical basis for feature misalignment in convolutional networks.

**Scale-Sensitivity of Convolutional Approaches.** In training classification models, an implicit assumption is that the strongest indicator of the class label is the target itself (i.e., the core regions). From Table 1, we observe through the significant influence of non-core regions that this assumption does not universally hold. In cases where the target is far from the camera, as commonly observed in Hard-ImageNet, the emphasis is placed on **learning the best non-core surrogate to the actual target**, rather than obtaining an accurate feature representation using just the fewer relevant regions.

Learning a non-core surrogate does reduce cross-entropy loss, but at the cost of misrepresenting the underlying classification target, thus inducing feature misalignment. The model should, through the course of training, distinguish and ignore non-core regions in determining the final prediction.

This leads us to propose an alignment-motivated constraint to empirical risk minimization.

**Definition 4.4** (Core-Constrained Risk Minimization).

$$\mathcal{R}_{\mathrm{CCRM}}(f) := \mathbb{E}_{(\mathbf{X},(H,\mathbf{y}))\sim\mathcal{D}} \left[ \ell(\tilde{f}(\mathbf{X}), \mathbf{y}) \right] \quad \text{s.t.} \quad \sum_{c=1}^{C} \left\| (1-H) \odot \mathbf{CAM}^{\mathrm{Cntrst}}_{(c_t,c)} \right\| = 0 \tag{14}$$

Where $\ell(f(\mathbf{X}), \mathbf{y}) = \mathbb{1}(\arg\max(\tilde{f}(\mathbf{X})) \neq \arg\max(\mathbf{y}))$ is 0/1 loss for the multiclass setting.

## 4.2 Core-Focused Cross-Entropy

We have shown that cross-entropy motivates generating predictions using either core or non-core features. To correct this, we propose Core-Focused Cross-Entropy, which penalizes the contribution from non-core regions to the final classification.

**Definition 4.5** (Core-Focused Cross-Entropy). We integrate masked region suppression to the definition of cross-entropy using the following formulation:

$$\mathcal{L}_{\text{CFCE}}(f(\mathbf{X}), \mathbf{y}, H) := \log \left( \sum_{c=1}^{C} \exp \left( -\sum H \odot \mathbf{CAM}_{(c_t,c)}^{\text{Cntrst}} + \sum (1-H) \odot |\mathbf{CAM}_{(c_t,c)}^{\text{Cntrst}}| \right) \right) \tag{15}$$

We can show that the above loss function is consistent with our constrained optimization objective.

**Theorem 4.6** (Consistency of Core-Focused Cross-Entropy). *A sequence of predictors $f_n$ that converges to the optimal $\mathcal{R}_{\text{CFCE}}$-risk also converges to the Bayes-optimal $\mathcal{R}_{\text{CCRM}}$-risk. Equivalently, in the realizable setting, $\mathcal{L}_{\text{CFCE}}$ is classification-calibrated.*

$$\mathcal{R}_{\text{CFCE}}(f_n) \to \mathcal{R}_{\text{CFCE}}^* \implies \mathcal{R}_{\text{CCRM}}(f_n) \to \mathcal{R}_{\text{CCRM}}^* \tag{16}$$

*Where $\mathcal{R}_{\text{CFCE}}(f)$ is defined as:*

$$\mathcal{R}_{\text{CFCE}}(f) := \mathbb{E}_{(\mathbf{X},(H,\mathbf{y}))\sim\mathcal{D}} \left[ \mathcal{L}_{\text{CFCE}}(f(\mathbf{X}), \mathbf{y}, H) \right] \tag{17}$$

**Divergence Regularization.** Using ContrastiveCAMs, we observe a tendency for cross-entropy to only generate contrast in regions where feature differences are prominent within the training set. Successful test predictions rely on the prominence of the same set of differing features even if there exist subtleties in the training set that can be used to offer more nuanced classifications. We thus propose regularization by minimizing divergence between target mask $H$ and $\mathbf{CAM}_c^{\text{Cntrst}}$. This encourages contrast for every region in which the target is present, even when the difference is subtle.

**Definition 4.7** (Regularized Core-Focused Cross-Entropy). We regularize $\mathcal{L}_{\text{CFCE}}$ to encourage contrast over the entire target region using KL Divergence:

$$\mathcal{L}_{\text{RCFCE}}(f(\mathbf{X}), \mathbf{y}, H) := \mathcal{L}_{\text{CFCE}} + \frac{\lambda_1}{C-1} \sum_{c \in [C] \backslash c_t} D_{KL} \left( \sigma(\lambda_2 H) \,||\, \sigma \left( \lambda_3 \mathbf{CAM}_{(c_t,c)}^{\text{Cntrst}} \right) \right) \tag{18}$$

The divergence term motivates similarity in the *shape* of ContrastiveCAMs to $H$. The normalizing behavior of softmax, analogous to its effect on the logits, means that absolute scale is invariant; that information comes exclusively from $\mathcal{L}_{\text{CFCE}}$.

Supplemental formulations and adaptations of core-focused optimization are deferred to Appendix B.

## 5 Experiments

For our experiments, we evaluate the performance of ResNet-50 with a set of interpretability-motivated modifications. These are detailed in Appendix C. For consistency, we include baselines with (denoted by 'w/ Arch') and without these modifications. We initialize each training run on ImageNet pre-trained weights, and report fine-tuning performance.

**Datasets.** We present training results for Oxford IIIT-Pets (Parkhi et al., 2012), Hard-ImageNet (Moayeri et al., 2022), and the Semantic Boundaries Dataset (Hariharan et al., 2011). These datasets span image classification tasks with binary, multiclass & multilabel targets. In addition to reporting raw prediction performance, we also report intersection-over-union (IoU) scores, indicating the overlap between ground-truth core regions and those used by the models for classification.

### 5.1 Hard-Imagenet

Hard-ImageNet (Moayeri et al., 2022) is a subset of ImageNet (Deng et al., 2009) that only contains classes that have been observed to use spurious features to inform predictions (Singla and Feizi, 2022).

The core regions from these classes typically constitute a minority of the overall image (13.96% on average), lending further evidence to the scale-sensitivity of convolutional models (Section 4.1).

To evaluate the performance of models using core regions only, Moayeri et al. (2022) introduces an evaluation suite that reports a) accuracy when core regions are removed from the image using segmentation masking, bounding-box masking and tiling over the foreground; b) *relative foreground sensitivity* (RFS) which evaluates performance degradation under corruption of the foreground; and c) saliency alignment measured by intersection over union of core masks to regions used for prediction.

Table 2: Hard-ImageNet benchmarks on finetuned ResNet-50 models trained using varying approaches. Models trained using our proposed core-focused loss functions show significant improvement across all evaluations, at the cost of some un-ablated performance.

| Method | Accuracy under Core-Region Ablation (%) | | | | RFS ($\uparrow$) | GradCAM IoU ($\uparrow$) | Contrastive-CAM IoU ($\uparrow$) |
|---|---|---|---|---|---|---|---|
| | None ($\uparrow$) | Gray Mask ($\downarrow$) | Gray BBOX ($\downarrow$) | Tile ($\downarrow$) | | | |
| Cross-Entropy | 94.25 | 75.94 | 69.39 | 67.38 | -0.18 | 18.44 | − |
| CORM (Singla et al., 2022) | 92.91 | 76.20 | 69.12 | 68.32 | -0.08 | 20.43 | − |
| DFR (Kirichenko et al., 2022) | **94.39** | 73.53 | 67.51 | 66.71 | -0.27 | 18.39 | − |
| CORM + DFR | 91.31 | 72.59 | 63.64 | 63.90 | -0.23 | 20.35 | − |
| CE w/ Arch | $93.69_{\pm0.77}$ | $76.53_{\pm2.15}$ | $72.49_{\pm2.19}$ | $71.02_{\pm2.4}$ | $-0.23_{\pm0.05}$ | $16.25_{\pm14.07}$ | $30.27_{\pm3.99}$ |
| CFCE (Ours) | $90.53_{\pm0.69}$ | $\mathbf{41.78}_{\pm\mathbf{1.49}}$ | $\mathbf{31.66}_{\pm\mathbf{1.26}}$ | $\mathbf{34.31}_{\pm\mathbf{1.04}}$ | $.224_{\pm0.10}$ | $18.88_{\pm1.13}$ | $89.22_{\pm0.31}$ |
| CFCE + KL (Ours) | $90.35_{\pm1.59}$ | $45.49_{\pm5.15}$ | $37.07_{\pm4.57}$ | $39.47_{\pm4.12}$ | $\mathbf{.236}_{\pm\mathbf{0.10}}$ | $\mathbf{51.52}_{\pm\mathbf{1.07}}$ | $\mathbf{93.39}_{\pm\mathbf{0.11}}$ |

IoU for this benchmark was computed using GradCAMs (Selvaraju et al., 2017) only for consistency with baselines, as GradCAMs have been shown to present unfaithful explanations (Draelos and Carin, 2020). We thus include additional evaluations using ContrastiveCAMs for core-focused models. We also qualitatively evaluate improvements using core-focused approaches in Figure 3 below.

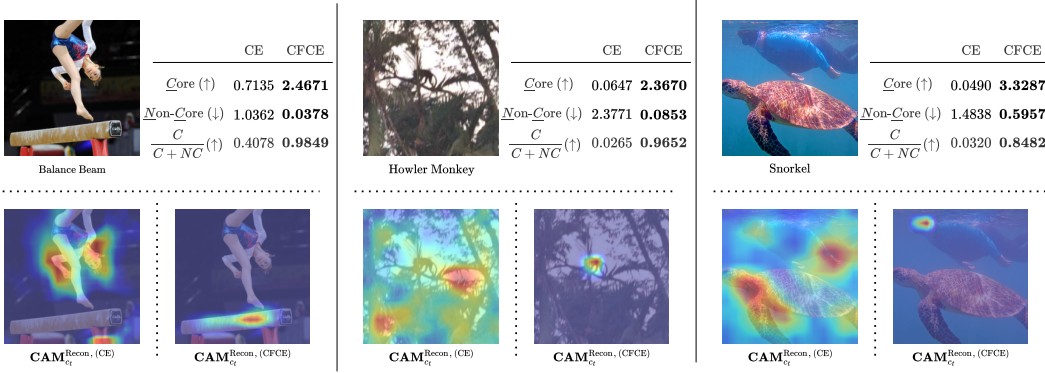

Figure 3: Models trained using CFCE exhibit suppressed contributions from non-core regions.

## 5.2 OXFORD IIIT-PETS

The Oxford IIIT-Pets dataset contains images of 37 breeds of cats and dogs, paired with segmentation trimaps that denote the foreground and background regions within the image. In the binary setting, the objective is to classify cats and dogs; individual breed labels are merged. This creates a class imbalance (4978 dogs to 2371 cats), however no training modifications are made to account for this. There is virtually no class imbalance in the multiclass setting.

**Applicability of Approximate Masks.** Core-region masks $H$ have a smaller resolution compared to input $\mathbf{X}$ as a consequence of the convolutional backbone $g$. Thus, in the absence of ground-truth core-region masks, approximate pixel-level masks or weaker supervision such as bounding boxes can be used to effectively suppress contributions from non-core regions. We demonstrate this empirically through competitive alignment achieved both with auto-generated masks obtained using Segment Anything (Kirillov et al., 2023) (SAM), and with weaker supervision via bounding boxes (BBOX).

| Method | Core Region Masks | Binary | | | | Multiclass | | | |
|---|---|---|---|---|---|---|---|---|---|
| | | Accuracy (%) | | IoU (%) | | Accuracy (%) | | IoU (%) | |
| | | Train | Valid | Train | Valid | Train | Valid | Train | Valid |
| Cross-Entropy | − | $99.82_{\pm0.26}$ | $99.40_{\pm0.07}$ | $78.37_{\pm1.12}$ | $78.37_{\pm1.14}$ | $99.92_{\pm0.21}$ | $94.41_{\pm1.07}$ | $80.04_{\pm0.66}$ | $80.16_{\pm0.48}$ |
| CE w/ Arch | − | $99.99_{\pm0.02}$ | $99.4_{\pm0.22}$ | $38.58_{\pm16.95}$ | $39.07_{\pm16.98}$ | $100_{\pm0}$ | $95.3_{\pm0.3}$ | $59.86_{\pm17.09}$ | $60.6_{\pm17.2}$ |
| CFCE | GT | $99.88_{\pm0.10}$ | $99.32_{\pm0.25}$ | $83.22_{\pm1.13}$ | $82.92_{\pm1.18}$ | $99.96_{\pm0.03}$ | $92.96_{\pm0.15}$ | $87.93_{\pm0.24}$ | $88.16_{\pm0.33}$ |
| CFCE + KL | GT | $99.71_{\pm0.27}$ | $99.32_{\pm0.15}$ | $\mathbf{94.93_{\pm0.88}}$ | $\mathbf{92.72_{\pm0.73}}$ | $99.74_{\pm0.13}$ | $90.08_{\pm1.47}$ | $\mathbf{96.22_{\pm3.58}}$ | $\mathbf{93.12_{\pm2.22}}$ |
| CFCE | SAM | $99.92_{\pm0.06}$ | $99.37_{\pm0.15}$ | $83.96_{\pm2.1}$ | $83.95_{\pm2.33}$ | $99.6_{\pm0.19}$ | $93.26_{\pm0.67}$ | $84.79_{\pm1.26}$ | $85.26_{\pm1.22}$ |
| CFCE + KL | SAM | $99.88_{\pm0.07}$ | $99.19_{\pm0.24}$ | $83.46_{\pm1.73}$ | $83.54_{\pm1.96}$ | $99.6_{\pm0.2}$ | $93.7_{\pm0.28}$ | $84.67_{\pm1.16}$ | $85.16_{\pm1.2}$ |
| CFCE | BBOX | $\mathbf{100_{\pm0.01}}$ | $\mathbf{99.42_{\pm0.22}}$ | $79.09_{\pm2.26}$ | $79.13_{\pm2.28}$ | $\mathbf{99.98_{\pm0}}$ | $93.83_{\pm0.33}$ | $84.26_{\pm1.86}$ | $84.61_{\pm1.91}$ |

Notably, KL regularization must not be applied when bounding boxes are used in place of masks, as fitting to the shape of the box mischaracterizes the target. Also note that ground-truth (GT) masks are used for validation in every setting to ensure a fair comparison.

## 5.3 SEMANTIC BOUNDARIES DATASET (PASCAL VOC)

The Semantic Boundaries Dataset introduces segmentation annotations to the entire Pascal VOC 2011 Dataset (Everingham et al., 2011). We use this dataset to demonstrate performance improvements for both classification and downstream detection settings.

**Classification.** PASCAL VOC encodes a 20-class *multilabel* classification task; thus input image may contain multiple positive classifications. We report a pareto improvement with increased Average Precision (AP) and Intersection-over-Union (IoU) scores when using core-focused loss formulations.

| Method | AP (%) | | IoU (%) | |
|---|---|---|---|---|
| | Train | Valid | Train | Valid |
| Cross-Entropy | $\mathbf{99.75}_{\pm\mathbf{0.30}}$ | $87.32_{\pm2.58}$ | $46.08_{\pm16.54}$ | $44.50_{\pm16.57}$ |
| CE w/ Arch | $99.57_{\pm0.74}$ | $\mathbf{88.85}_{\pm\mathbf{0.79}}$ | $40.69_{\pm16.37}$ | $38.55_{\pm16.43}$ |
| CFBCE | $98.38_{\pm2.49}$ | $88.39_{\pm1.23}$ | $85.00_{\pm1.32}$ | $82.07_{\pm0.91}$ |
| CFBCE + KL | $97.92_{\pm1.00}$ | $87.19_{\pm0.46}$ | $\mathbf{89.53}_{\pm\mathbf{1.89}}$ | $\mathbf{85.39}_{\pm\mathbf{0.60}}$ |

**Segmentation.** We also report improvements in IoU performance of core-focused backbones on downstream segmentation, both when fine-tuned (i.e., with a frozen backbone) and trained end-to-end.

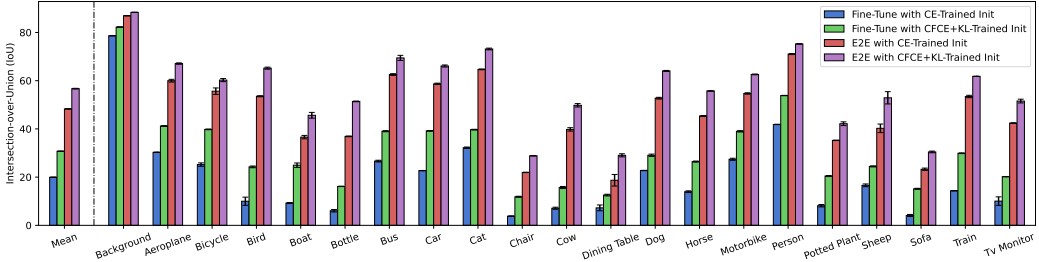

## 6 DISCUSSION

In this work, we establish a connection between interpretability and feature alignment. We demonstrate the impact of utilizing *post-hoc* (i.e., post-training) explainability methods, primarily used as sanity checks, as a guiding factor during training to improve feature alignment with encouraging effect. Core-Focused Cross Entropy is a direct result of the desirable theoretical properties of ContrastiveCAMs, establishing the value of correctness guarantees in interpretability. Reductive metrics inevitably present a partial view of factors that influence model prediction, and comprehensively ensuring that deep neural networks faithfully learn to solve the intended, underlying objective remains a significant challenge for the research community. We hope that our work motivates further exploration towards connections between interpretability and alignment of deep neural networks.

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

## A  MATHEMATICAL DERIVATIONS

**Proposition 3.1.** *The softmax function is invariant to a universal shift of all its input components:*

$$\sigma(\mathbf{x}) = \sigma(\mathbf{x} + a\mathbf{1}_C) \quad \forall \mathbf{x} \in \mathbb{R}^C,\ a \in \mathbb{R}$$

*Proof.*

$$\sigma(\mathbf{x} + a\mathbf{1}_C) = \frac{1}{\sum_{c=1}^{C} e^{\mathbf{x}_c + a}}(e^{\mathbf{x}_1 + a}, e^{\mathbf{x}_2 + a}, \cdots, e^{\mathbf{x}_C + a}) = \frac{1}{\sum_{c=1}^{C} e^{\mathbf{x}_c}}(e^{\mathbf{x}_1}, e^{\mathbf{x}_2}, \cdots, e^{\mathbf{x}_C}) = \sigma(\mathbf{x}). \tag{19}$$

$\square$

**Theorem 3.2.** *HiResCAM explanations* $\mathbf{CAM}^{\mathrm{HiRes}} \in \mathbb{R}^{C \times d_1 \times d_2}$ *corresponding to probability predictions* $\tilde{f}(\mathbf{X}) \in \mathbb{R}^C$ *are not uniquely determined, admitting a universal shift of class-level explanations* $\mathbf{CAM}_c^{\mathrm{HiRes}}$ *by an arbitrary matrix* $M \in \mathbb{R}^{d_1 \times d_2} \ \forall c \in [C]$.

$$\tilde{f}(\mathbf{X}) = \sigma\left(\sum_{i=1, j=1}^{d_1, d_2} \mathbf{CAM}_{:,i,j}^{\mathrm{HiRes}} + \mathbf{b}\right) = \sigma\left(\sum_{i=1, j=1}^{d_1, d_2} \overline{\mathbf{CAM}}_{:,i,j}^{\mathrm{HiRes}} + \mathbf{b}\right) \quad \forall M \in \mathbb{R}^{d_1 \times d_2} \tag{20}$$

*Where* $\overline{\mathbf{CAM}}^{\mathrm{HiRes}}$ *is defined as:*

$$\overline{\mathbf{CAM}}_c^{\mathrm{HiRes}} := \mathbf{CAM}_c^{\mathrm{HiRes}} + M \qquad \forall c \in [C] \tag{21}$$

*Proof.* First, we define the set of all valid shifts $\mathcal{M}$:

$$\text{Let } \mathcal{M} := \{\mathbf{M} \in \mathbb{R}^{C \times d_1 \times d_2} : \mathbf{M}_i = \mathbf{M}_j \quad \forall i, j \in [C]\} \tag{22}$$

The matrix $\mathbf{M}_i \in \mathbb{R}^{d_1 \times d_2}$ can be arbitrary, provided it is constant $\forall i \in [C]$. Thus $|\mathcal{M}| = \infty$. We will show that all HiResCAM explanations that differ by $\mathbf{M} \in \mathcal{M}$ form an equivalence class under the *softmax* operation. Consider the following set:

$$[\overline{\mathbf{CAM}}^{\text{HiRes}}] = \{\mathbf{CAM}^{\text{HiRes}} + \mathbf{M} : \mathbf{M} \in \mathcal{M}\} \tag{23}$$

We then show that any $\overline{\mathbf{CAM}}^{\text{HiRes}}$ with a corresponding shift $\mathbf{M}'$ is a valid explanation (i.e., preserves the final prediction). With logits $f$ deconstructed into HiResCAMs following Eq. (3), we have:

$$\sigma \left( \sum_{i=1, j=1}^{d_1, d_2} \overline{\mathbf{CAM}}_{:,i,j}^{\text{HiRes}} + \mathbf{b} \right) = \sigma \left( \sum_{i=1, j=1}^{d_1, d_2} \left( \overline{\mathbf{CAM}}_{:,i,j}^{\text{HiRes}} + \mathbf{M}'_{:,i,j} \right) + \mathbf{b} \right) \tag{24}$$

Let $a = \sum_{i=1, j=1}^{d_1, d_2} \mathbf{M}'_{c,i,j}$ for some $c \in [C]$. By property of $\mathbf{M}$:

$$= \sigma \left( \sum_{i=1, j=1}^{d_1, d_2} \left( \mathbf{CAM}_{:,i,j}^{\text{HiRes}} \right) + a\mathbf{1}_C + \mathbf{b} \right) \tag{25}$$

Applying Proposition 3.1, we have:

$$= \sigma \left( \sum_{i=1, j=1}^{d_1, d_2} \mathbf{CAM}_{:,i,j}^{\text{HiRes}} + \mathbf{b} \right) = \tilde{f}(\mathbf{X}) \tag{26}$$

Thus, we have:

$$\tilde{f}(\mathbf{X}) = \sigma \left( \sum_{i=1, j=1}^{d_1, d_2} \mathbf{CAM}_{:,i,j}^{\text{HiRes}} + \mathbf{b} \right) = \sigma \left( \sum_{i=1, j=1}^{d_1, d_2} \overline{\mathbf{CAM}}_{:,i,j}^{\text{HiRes}} + \mathbf{b} \right) \quad \forall M \in \mathbb{R}^{d_1 \times d_2} \tag{27}$$

Proving the desired statement. $\qquad \square$

**Theorem 3.5.** *Let* $\mathbf{CAM}^{\text{HiRes}}$ *and* $\overline{\mathbf{CAM}}^{\text{HiRes}}$ *be two HiResCAMs corresponding to probability predictions* $\tilde{f}(\mathbf{X}) \in \mathbb{R}^C$ *such that:*

$$\overline{\mathbf{CAM}}^{\text{HiRes}} = \left\{ \mathbf{CAM}_c^{\text{HiRes}} + M : c \in [C] \right\} \tag{28}$$

*Then, for every* $M \in \mathbb{R}^{d_1 \times d_2}$*, it holds that:*

$$\mathbf{CAM}_{c_t}^{\text{Cntrst}} = \overline{\mathbf{CAM}}_{c_t}^{\text{Cntrst}} \quad \text{and} \quad \mathbf{CAM}^{\text{Recon}} = \overline{\mathbf{CAM}}^{\text{Recon}} \tag{29}$$

*Proof.* For some $c_t \in [C]$, we have:

$$\overline{\mathbf{CAM}}_{c_t}^{\text{Cntrst}} = \left\{ \overline{\mathbf{CAM}}_{(c_t, c)}^{\text{Cntrst}} : c \in [C] \setminus c_t \right\} \tag{30}$$

$$= \left\{ \overline{\mathbf{CAM}}_{c_t}^{\text{HiRes}} - \overline{\mathbf{CAM}}_c^{\text{HiRes}} : c \in [C] \setminus c_t \right\} \tag{31}$$

By definition of $\overline{\mathbf{CAM}}^{\text{HiRes}}$, we have:

$$= \left\{ \mathbf{CAM}_{c_t}^{\text{HiRes}} + M - \mathbf{CAM}_c^{\text{HiRes}} - M : c \in [C] \setminus c_t \right\} \tag{32}$$

$$= \left\{ \mathbf{CAM}_{c_t}^{\text{HiRes}} - \mathbf{CAM}_c^{\text{HiRes}} : c \in [C] \setminus c_t \right\} \tag{33}$$

$$= \left\{ \mathbf{CAM}_{(c_t, c)}^{\text{Cntrst}} : c \in [C] \setminus c_t \right\} = \mathbf{CAM}_{c_t}^{\text{Cntrst}} \tag{34}$$

$$\therefore \mathbf{CAM}_{c_t}^{\mathrm{Cntrst}} = \overline{\mathbf{CAM}}_{c_t}^{\mathrm{Cntrst}} \tag{35}$$

This proves the first statement. Now, we can tend to the $\mathbf{CAM}^{\mathrm{Recon}}$ case:

$$\overline{\mathbf{CAM}}_{c_t}^{\mathrm{Recon}} = \overline{\mathbf{CAM}}_{c_t}^{\mathrm{HiRes}} - \frac{1}{C}\sum_{c=1}^{C}\overline{\mathbf{CAM}}_c^{\mathrm{HiRes}} \tag{36}$$

By definition of $\overline{\mathbf{CAM}}^{\mathrm{HiRes}}$, we have:

$$= \mathbf{CAM}_{c_t}^{\mathrm{HiRes}} + M - \frac{1}{C}\sum_{c=1}^{C}\left(\mathbf{CAM}_c^{\mathrm{HiRes}} + M\right) \tag{37}$$

$$= \mathbf{CAM}_{c_t}^{\mathrm{HiRes}} + M - \frac{C \cdot M}{C} - \frac{1}{C}\sum_{c=1}^{C}\mathbf{CAM}_c^{\mathrm{HiRes}} \tag{38}$$

$$= \mathbf{CAM}_{c_t}^{\mathrm{HiRes}} - \frac{1}{C}\sum_{c=1}^{C}\mathbf{CAM}_c^{\mathrm{HiRes}} = \mathbf{CAM}_{c_t}^{\mathrm{Recon}} \tag{39}$$

$$\therefore \mathbf{CAM}^{\mathrm{Recon}} = \overline{\mathbf{CAM}}^{\mathrm{Recon}} \tag{40}$$

Proving the desired statements. $\square$

**Proposition 4.1.** *Softmax-activated class probabilities $\tilde{f}$ can be expressed as a direct function of ContrastiveCAMs and the bias vector.*

$$\tilde{f}_{c_t}(\mathbf{X}) = \left(\sum_{c=1}^{C}\exp\left(\mathbf{b}_c - \mathbf{b}_{c_t} - \sum\mathbf{CAM}_{(c_t,c)}^{\mathrm{Cntrst}}\right)\right)^{-1} \qquad \forall c_t \in [C] \tag{41}$$

*Where $\mathbf{CAM}_{(c_t,c_t)}^{\mathrm{Cntrst}} = \mathbf{0}_{d_1 \times d_2}$.*

*Proof.* Individual class probabilities for logit vector $f$ are defined as:

$$\tilde{f}_{c_t} = \sigma_{c_t}(f) = \frac{e^{f_{c_t}}}{\sum_i e^{f_i}} \tag{42}$$

For some $c_t \in [C]$.

We define our logit vector in terms of the elementwise difference to a target class $c$:

$$\mathbf{d} := f - f_{c_t} \implies f = f_{c_t} + \mathbf{d} \tag{43}$$

Based on this definition, class probabilities can equivalently be computed as:

$$\tilde{f}_{c_t} = \frac{e^{f_{c_t}}}{\sum_i e^{f_i}} = \frac{e^{f_{c_t}}}{\sum_i e^{f_{c_t}+\mathbf{d}_i}} = \frac{e^{f_{c_t}}}{e^{f_{c_t}}\sum_i e^{\mathbf{d}_i}} = \frac{1}{\sum_i e^{\mathbf{d}_i}} \tag{44}$$

This re-contextualizes softmax as a direct function of the differences of class logits. We can further deconstruct the difference by logit values:

$$\mathbf{d}_c = f_c - f_{c_t} = \sum_{i=1,j=1}^{d_1,d_2}\mathbf{CAM}_{c,i,j}^{\mathrm{HiRes}} + \mathbf{b}_c - \sum_{i=1,j=1}^{d_1,d_2}\mathbf{CAM}_{c_t,i,j}^{\mathrm{HiRes}} - \mathbf{b}_{c_t} \tag{45}$$

Applying Definition 3.3, we have:

$$\mathbf{d}_c = \mathbf{b}_c - \mathbf{b}_{c_t} - \sum\mathbf{CAM}_{(c_t,c)}^{\mathrm{Cntrst}} \tag{46}$$

Substituting $\mathbf{d}_i$ from Eq. (46) into Eq. (44), we have:

$$\tilde{f}_{c_t}(\mathbf{X}) = \frac{1}{\sum_{i=1}^{C}\exp\left(\mathbf{b}_c - \mathbf{b}_{c_t} - \sum\mathbf{CAM}_{(c_t,c)}^{\mathrm{Cntrst}}\right)} = \left(\sum_{c=1}^{C}\exp\left(\mathbf{b}_c - \mathbf{b}_{c_t} - \sum\mathbf{CAM}_{(c_t,c)}^{\mathrm{Cntrst}}\right)\right)^{-1} \tag{47}$$

We can thus compute class probabilities as a direct function of ContrastiveCAMs and the bias vector. $\square$

**Proposition 4.2.** *Given bias-free classifier h, we can precisely associate the impact of specific regions, encoded by binary mask H, to the computation of cross-entropy loss.*

$$\mathcal{L}_{\text{CE}}(f(\mathbf{X}), \mathbf{y}, H) = \log \left( \sum_{c=1}^{C} \exp \left( -\sum H \odot \mathbf{CAM}_{(c_t,c)}^{\text{Cntrst}} - \sum (1-H) \odot \mathbf{CAM}_{(c_t,c)}^{\text{Cntrst}} \right) \right) \tag{48}$$

*Proof.* Setting $\mathbf{b} = 0$ to the result from Proposition 4.1, we have:

$$\tilde{f}_{c_t}(\mathbf{X}) = \left( \sum_{c=1}^{C} \exp \left( -\sum \mathbf{CAM}_{(c_t,c)}^{\text{Cntrst}} \right) \right)^{-1} \tag{49}$$

For target class $c_t \in [C]$. Let $H$ and $(1-H)$ define core and non-core masks respectively; these are disjoint. We can use this to further disassociate ContrastiveCAMs:

$$\tilde{f}_{c_t} = \left( \sum_{c=1}^{C} \exp \left( -\sum H \odot \mathbf{CAM}_{(c_t,c)}^{\text{Cntrst}} - \sum (1-H) \odot \mathbf{CAM}_{(c_t,c)}^{\text{Cntrst}} \right) \right)^{-1} \tag{50}$$

For one-hot encoded target vector $\mathbf{y}$ and target class index $c_t$, cross-entropy loss is defined as:

$$\mathcal{L}_{CE}(f(\mathbf{X}), \mathbf{y}, H) = -\sum_{c=1}^{C} \mathbf{y}_c \log \tilde{f}_c = -\log \tilde{f}_{c_t} \tag{51}$$

To which we can substitute softmax using Eq. (50):

$$\mathcal{L}_{CE}(f(\mathbf{X}), \mathbf{y}, H) = -\log \left( \sum_{c=1}^{C} \exp \left( -\sum H \odot \mathbf{CAM}_{(c_t,c)}^{\text{Cntrst}} - \sum (1-H) \odot \mathbf{CAM}_{(c_t,c)}^{\text{Cntrst}} \right) \right)^{-1}$$

$$= \log \left( \sum_{c=1}^{C} \exp \left( -\sum H \odot \mathbf{CAM}_{(c_t,c)}^{\text{Cntrst}} - \sum (1-H) \odot \mathbf{CAM}_{(c_t,c)}^{\text{Cntrst}} \right) \right) \tag{52}$$

As core and non-core masks are disjoint, Eq. (52) enables us to identify the logit contributions from the core and non-core regions respectively. $\qquad\square$

**Theorem 4.6.** *A sequence of predictors $f_n \subset \mathcal{F}$ that converges to the optimal $\mathcal{R}_{\text{CFCE}}$-risk also converges to the Bayes-optimal $\mathcal{R}_{\text{CCRM}}$-risk. Equivalently, in the realizable setting, $\mathcal{L}_{\text{CFCE}}$ is classification-calibrated.*

$$\mathcal{R}_{\text{CFCE}}(f_n) \to \mathcal{R}_{\text{CFCE}}^* \implies \mathcal{R}_{\text{CCRM}}(f_n) \to \mathcal{R}_{\text{CCRM}}^* \tag{53}$$

*Where $\mathcal{R}_{\text{CFCE}}(f)$ is:*

$$\mathcal{R}_{\text{CFCE}}(f) := \mathbb{E}_{(\mathbf{X},(H,\mathbf{y}))\sim\mathcal{D}} \left[ \mathcal{L}_{\text{CFCE}}(f(\mathbf{X}), \mathbf{y}, H) \right] \tag{54}$$

*Proof.* We start by restating Definition (4.5):

$$\mathcal{L}_{\text{CFCE}}(f(\mathbf{X}), \mathbf{y}, H)) = \log \Bigg( \sum_{c=1}^{C} \exp \Big( -\sum H \odot \mathbf{CAM}_{(c_t,c)}^{\text{Cntrst}}$$

$$+ \sum (1-H) \odot |\mathbf{CAM}_{(c_t,c)}^{\text{Cntrst}}| \Big) \Bigg) \tag{55}$$

$$= \log \left( \sum_{c=1}^{C} \frac{\exp \left( \sum (1-H) \odot |\mathbf{CAM}_{(c_t,c)}^{\text{Cntrst}}| \right)}{\exp \left( \sum H \odot \mathbf{CAM}_{(c_t,c)}^{\text{Cntrst}} \right)} \right) \tag{56}$$

We can observe that $\mathcal{R}_{\mathrm{CFCE}}(f)$ takes the following form:

$$\mathcal{R}_{\mathrm{CFCE}}(f) = \mathbb{E}_{(\mathbf{X},(H,\mathbf{y})\sim\mathcal{D}}\left[\log\left(\sum_{c=1}^{C}\underbrace{\frac{\exp\left(\sum(1-H)\odot|\mathbf{CAM}_{(c_t,c)}^{\mathrm{Cntrst}}|\right)}{\exp\left(\sum H\odot\mathbf{CAM}_{(c_t,c)}^{\mathrm{Cntrst}}\right)}}_{s_c}\right)\right] \tag{57}$$

$\mathcal{R}_{\mathrm{CFCE}}^{*} = \inf_f \mathcal{R}_{\mathrm{CFCE}}(f)$ is predicated on each summand $s_c \to 0$. We have that:

$$\inf_f\left(\sum_{c=1}^{C}\frac{\exp\left(\sum(1-H)\odot|\mathbf{CAM}_{(c_t,c)}^{\mathrm{Cntrst}}|\right)}{\exp\left(\sum H\odot\mathbf{CAM}_{(c_t,c)}^{\mathrm{Cntrst}}\right)}\right) \geq \sum_{c=1}^{C}\frac{\inf_f\left(\exp\left(\sum(1-H)\odot|\mathbf{CAM}_{(c_t,c)}^{\mathrm{Cntrst}}|\right)\right)}{\sup_f\left(\exp\left(\sum H\odot\mathbf{CAM}_{(c_t,c)}^{\mathrm{Cntrst}}\right)\right)} \tag{58}$$

Given sufficiently expressive $\mathcal{F}$ by assumption of realizability of $\mathcal{R}_{\mathrm{CCRM}}^{*}$, as $n\to\infty$, $f_n$ converges uniformly towards the equality case thus admitting the following dual objective for each $s_c$:

$$\mathcal{R}_{\mathrm{CFCE}}(f_n) \to \mathcal{R}_{\mathrm{CFCE}}^{*} \iff \frac{\inf_f\left(\exp\left(\sum(1-H)\odot|\mathbf{CAM}_{(c_t,c)}^{\mathrm{Cntrst}}|\right)\right)}{\sup_f\left(\exp\left(\sum H\odot\mathbf{CAM}_{(c_t,c)}^{\mathrm{Cntrst}}\right)\right)} \quad \forall c\in[C] \tag{59}$$

With the absolute $|\cdot|$ operator over numerator's exponent and the realizability assumption, we have:

$$\inf_f\left(\exp\left(\sum(1-H)\odot|\mathbf{CAM}_{(c_t,c)}^{\mathrm{Cntrst}}|\right)\right) = 1 \iff \|(1-H)\odot\mathbf{CAM}_{(c_t,c')}^{\mathrm{Cntrst}}\| = 0 \tag{60}$$

This satisfies the constraint from Definition 4.4 and further implies (by absolute homogeneity of the norm) that each non-core region has no contribution to the final classification.

Next, we can tend to the denominator.

$$\text{Let } f^* = \arg\sup_f\left(\sum H\odot\mathbf{CAM}_{(c_t,c)}^{\mathrm{Cntrst}}\right) \tag{61}$$

By convexity of $\exp$, we have that:

$$\exp\left(\sum H\odot\mathbf{CAM}_{(c_t,c)}^{\mathrm{Cntrst},f^*}\right) \geq \sup_f\left(\exp\left(\sum H\odot\mathbf{CAM}_{(c_t,c)}^{\mathrm{Cntrst}}\right)\right) \tag{62}$$

The realization of $f^*$ satisfies the following condition:

$$\sum H\odot\mathbf{CAM}_{(c_t,c)}^{\mathrm{Cntrst}} > 0 \qquad \forall c\in[C] \tag{63}$$

Which is sufficient to show the largest logit is that of the target class $c_t$. Thus $\arg\max(f(\mathbf{X})) = \arg\max(\mathbf{y}) \ \forall(\mathbf{X},(H,\mathbf{y}))\sim\mathcal{D} \implies \mathbb{E}_{(\mathbf{X},(H,\mathbf{y}))\sim\mathcal{D}}[\ell(f(\mathbf{X},\mathbf{y}))] = 0$ which gives us:

$$\mathcal{R}_{\mathrm{CFCE}}(f_n) \to \mathcal{R}_{\mathrm{CFCE}}^{*} \implies \mathcal{R}_{\mathrm{CCRM}}(f_n) \to \mathcal{R}_{\mathrm{CCRM}}^{*} \tag{64}$$

Proving the consistency of $\mathcal{L}_{\mathrm{CFCE}}$ as a surrogate minimizer to $\mathcal{R}_{\mathrm{CCRM}}$. $\qquad\square$

**Proposition B.1.** *We can integrate background suppression to the definition of binary cross-entropy using the following formulation:*

$$\mathcal{L}_{\mathrm{CFBCE}}(f(\mathbf{X}),\mathbf{y},H) = -\frac{1}{C}\sum_{i=1}^{C}\Bigg[\mathbf{y}_i\log\left(\phi\left(\sum_{j,k}H_i\odot\mathbf{CAM}_{i,j,k}^{\mathrm{HiRes}} - \sum_{j,k}(1-H_i)\odot|\mathbf{CAM}_{i,j,k}^{\mathrm{HiRes}}|\right)\right)$$
$$+ (1-\mathbf{y}_i)\log\left(1-\tilde{f}(\mathbf{X})_i\right)\Bigg] \tag{65}$$

*Proof.* We will prove for the multilabel setting, which is a generalization of binary cross-entropy. For binary vector $\mathbf{y}$ (i.e., $\mathbf{y}_i \in \{0, 1\} \; \forall i$), class-specific core masks $H_i$, and sigmoid $\phi$ activated logits $f$, denoted $\tilde{f}$, binary cross-entropy is defined as:

$$\mathcal{L}_{\text{BCE}}(f(\mathbf{X}), \mathbf{y}, H) = -\frac{1}{C} \sum_{i=1}^{C} \left[ \mathbf{y}_i \log \tilde{f}_i + (1 - \mathbf{y}_i) \log \left( 1 - \tilde{f}_i \right) \right] \tag{66}$$

$$= -\frac{1}{C} \sum_{i=1}^{C} \left[ \mathbf{y}_i \log \phi(f_i) + (1 - \mathbf{y}_i) \log \left( 1 - \phi(f_i) \right) \right] \tag{67}$$

Setting $\mathbf{b} = 0$, we can substitute Eq. (3) within the first term:

$$= -\frac{1}{C} \sum_{i=1}^{C} \left[ \mathbf{y}_i \log \phi \left( \sum_{j,k} \mathbf{CAM}_{i,j,k}^{\text{HiRes}} \right) + (1 - \mathbf{y}_i) \log \left( 1 - \tilde{f}_i \right) \right] \tag{68}$$

Similar to Proposition 4.5, we can break down each HiResCAM to core and spurious components. For non-target indices, we seek to reducing logit values across the entire input image. Therefore, we do not disassociate logit values for the second term.

$$\mathcal{L}_{\text{BCE}}(f(\mathbf{X}), \mathbf{y}, H) = -\frac{1}{C} \sum_{i=1}^{C} \left[ \mathbf{y}_i \log \phi \left( \sum_{j,k} H_i \odot \mathbf{CAM}_{i,j,k}^{\text{HiRes}} + \sum_{j,k} (1 - H_i) \odot \mathbf{CAM}_{i,j,k}^{\text{HiRes}} \right) \right.$$
$$\left. + (1 - \mathbf{y}_i) \log \left( 1 - \tilde{f}_i \right) \right] \tag{69}$$

The current formulation motivates activating either the core or non-core for positive classification, and motivates de-activating every pixel of the non-positive class. We penalize activation on the non-core regions for the positive class only:

$$\mathcal{L}_{\text{CFBCE}}(f(\mathbf{X}), \mathbf{y}, H) = -\frac{1}{C} \sum_{i=1}^{C} \left[ \mathbf{y}_i \log \left( \phi \left( \sum_{j,k} H_i \odot \mathbf{CAM}_{i,j,k}^{\text{HiRes}} - \sum_{j,k} (1 - H_i) \odot |\mathbf{CAM}_{i,j,k}^{\text{HiRes}}| \right) \right) \right.$$
$$\left. + (1 - \mathbf{y}_i) \log \left( 1 - \tilde{f}(\mathbf{X})_i \right) \right] \tag{70}$$

This gives us the core-focused binary cross-entropy formulation. $\qquad \square$

# B   CORE-FOCUSED CROSS-ENTROPIC ADAPTATIONS

## B.1   CORE-FOCUSED BINARY CROSS-ENTROPY

For sigmoid-activated binary / multilabel classification tasks, we leverage similar principles to define core-focused binary cross-entropy. Since we do not have the contrastive process in softmax-activation, this definitions relies only on HiResCAMs. We represent sigmoid activation using $\phi$ and admit $C$ target-region masks, denoted $H_i$ for each class $i \in [C]$. In addition, instead of one-hot encoding, we now have binary vector $\mathbf{y}$ (i.e., $\mathbf{y}_i \in \{0, 1\} \; \forall i$).

**Proposition B.1** (Core-Focused Binary Cross-Entropy)**.** *We can integrate background suppression to the definition of binary cross-entropy using the following formulation:*

$$\mathcal{L}_{\text{CFBCE}}(f(\mathbf{X}), \mathbf{y}, H) = -\frac{1}{C} \sum_{i=1}^{C} \left[ \mathbf{y}_i \log \left( \phi \left( \sum_{j,k} H_i \odot \mathbf{CAM}_{i,j,k}^{\text{HiRes}} - \sum_{j,k} (1 - H_i) \odot \left| \mathbf{CAM}_{i,j,k}^{\text{HiRes}} \right| \right) \right) \right.$$
$$\left. + (1 - \mathbf{y}_i) \log \left( 1 - \tilde{f}(\mathbf{X})_i \right) \right] \tag{71}$$

**Divergence Regularization**   Similar to Definition 4.5, we define a divergence term for the target class to motivate activation of the entire core region within the training objective.

**Definition B.2** (Regularized Core-Focused Binary Cross-Entropy)**.**

$$\mathcal{L}_{\text{RCFBCE}}(f(\mathbf{X}), \mathbf{y}, H) = \mathcal{L}_{\text{CFBCE}} + \frac{\lambda_1}{\|\mathbf{y}\|_1} \sum_{i=1}^{C} \mathbf{y}_i D_{\text{KL}} \left( \sigma(\lambda_2 H_i) \, || \, \sigma \left( \lambda_3 \mathbf{CAM}_i^{\text{HiRes}} \right) \right) \quad (72)$$

## B.2   CutMix with Core-Focused Cross-Entropy

CutMix (Yun et al., 2019) is a batch-wise augmentation technique that encourages better regularization by a) "cutting" out a randomized rectangle (randomized portion remaining consistent across the batch) of a given image and b) "mixing" the cut-out with it's neighbor. The corresponding labels are mixed by a randomly sampled parameter $\lambda$.

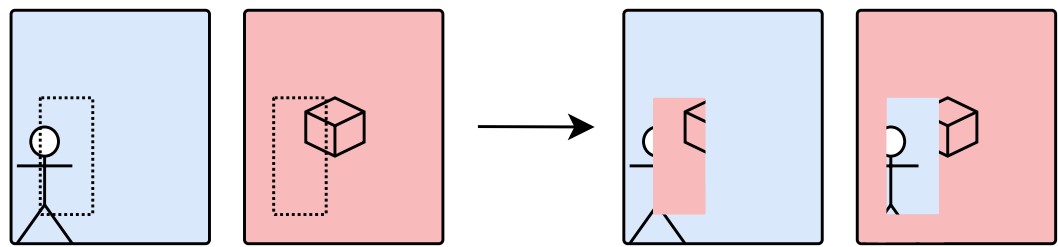

**Definition B.3** (CutMix with Core-Focused Cross-Entropy)**.**  Let segmentation mask $H$ take the following form:

$$H := \begin{cases} -1 & \text{pixel does not contain any class} \\ c & \text{pixel contains class } c \end{cases}$$

Also, let $\mathbb{1}_a$ be the indicator function applied elementwise for some $a \in \mathbb{R}$.

Then, Core-Focused Cross Entropy (4.5) with CutMix is formulated as follows:

$$\begin{aligned}
\mathcal{L}_{\text{CM\_CFBCE}}(f(\mathbf{X}), H, \mathbf{y}) = \log \Bigg( \sum_i \exp \Bigg( &- \sum \mathbb{1}_c(H) \odot \mathbf{CAM}_{(c,i)}^{\text{Cntrst}} \\
&+ \sum |\mathbb{1}_{-1}(H) \odot \mathbf{CAM}_{(c,i)}^{\text{Cntrst}}| + \sum \mathbb{1}_i(H) \odot \mathbf{CAM}_{(c,i)}^{\text{Cntrst}} \Bigg) \Bigg)
\end{aligned} \quad (73)$$

Where the third newly introduced term within the exponent expresses differential contrast.

## C   Training Details

**Hyperparameters.**   To *mitigate reward-hacking* our proposed approach, we selected a consistent set of hyperparameters that generally performs well and use it across all our experiments. We train each model using the Adam optimizer (Kingma, 2014) for 150 epochs with a learning rate of $5 \cdot 10^{-4}$, using a linear warmup of 5 epochs followed by Cosine Annealing (Loshchilov and Hutter, 2016) for the remaining 145 epochs. We use a weight decay of $10^{-4}$, a batch size of 768. For divergence regularized approaches, we used $\boldsymbol{\lambda} = \{50, \ 10^3, \ 10\}$.

**Reproducibility.**   The source code, datasets, experiments, evals, and model weights are published under a permissive license and can be found at *[redacted for double blind peer-review]*.

## C.1   Architecture Modifications

The architecture used for training was ResNet-50 (He et al., 2016), initialized with ImageNet. We introduce the following three key modifications:

**Removed final downsampling.** For images of size $(224, 224)$, the final downsampling layer converts the latent feature embeddings from $d_1 = d_2 = 14$ to 7. This prohibitively reduces the size of the activation map, and making it hard to capture relevant features. We replace the stride of the final downsampling convolution to $(1, 1)$, matching that of the definition used through the rest of ResNet.

**Removed final bias.** The bias vector $\mathbf{b}$ within $h$ is not involved in the computation of the class activation map. However, it does affect predictions in a way that is not explained by ContrastiveCAMs. To maintain faithfulness of the explanations, we omit the bias from the final model architecture.

**Removed final BatchNormalization & ReLU.** Since the HiResCAM construction establishes convolution followed immediately by GAP, the standard architecture which uses BatchNormalization & ReLU layers after each convolution, does not directly explain the class score. We therefore neutralize those functions for the final convolutional block. This recovers the faithfulness guarantee.

Note that the above changes correspond only to the final convolutional block of the backbone $g$ and the bias of the linear classifier $h$; the rest of the architecture remains consistent.

