# OpenReview forum: "Improving Feature Alignment in ConvNets using ContrastiveCAMs and Core-Focused Cross-Entropy"
_ICLR.cc/2026/Conference — Submitted to ICLR 2026_

### Official Review · Reviewer_BTYW · 2025-10-28

**Soundness:** 2
**Presentation:** 2
**Contribution:** 1
**Rating:** 2
**Confidence:** 3

**Summary:**

This paper proposes a interpretability framework to improve feature alignment in CNNs. The authors theoretically demonstrate that HiResCAM explanations are not unique because of the softmax activation, and adding a shift via a common matrix
 can distort attention maps. To solve this, they propose ContrastiveCAMs, which remove this redundancy by producing shift-invariant, attention maps that more accurately highlight discriminative image regions. Experiments show that models depend on non-core, irrelevant regions, specially when regions occupy small portions of images.They then develop Core-Focused Cross-Entropy, a modified loss function that suppresses attention to non-core areas while also making target features more relevant. The method performed good across multiclass and binary classification tasks and some segmentation benchmarks.

**Strengths:**

- The work is theoretically grounded. They theoretically show that HighResCAM explanations are not uniquely determined because of arbitrary shifts via softmax invariance, and provide formal proofs.
- The paper proposes a solution by connecting interpretability output with training objective to improve feature alignment. Such as actionable interpretability technique is impressive.

**Weaknesses:**

- This paper investigates CNNs, not ViTs. I think the whole field has shifted to ViTs long time ago. I am not saying CNNs are obsolete, but having ViT variants (along with CNN variants) are a must now. What about CLIP models?
- The paper investigates one (out of many) variation of CAM (here, HighResCAM ), and HighResCAM is not even a published work. I am not sure whether this problem is worth investigating in the first place. What about other CAM methods? What about other explanation methods (saliency-based)?
- The finding is not novel, in essence similar works have observed the same thing, see [R1]. This work is not even cited.
- There are many ways of explanation-guided learning. See [R2]. The authors did not consider these works in their comparisons. Is their method better to [R1, R2 methods]? Does it work on other explainability methods?
- There are no results on ImageNet. The authors report on Hard-ImageNet but not ImageNet. Does the method improve performance on the ImageNet?
- The introduction lacks motivation. It lists some works and their importances, and then jumps directly to "In this work, we develop...".
- The related work section provides some similar works but the most important part of that section is missing; the authors should clearly state the difference between the related works and their own work, how they tackle the problems and the limitations of those works, and why their method is better.


[R1] Consistent Explanations by Contrastive Learning
[R2] Studying How to Efficiently and Effectively Guide Models with Explanations

**Questions:**

I think this paper is not ready for ICLR due to the weaknesses mentioned. In particular, it investigates only CNNs, and a very special case (HighResCAM) of a very special case (CAM) of the wide variety of explainability methods. Furthermore, the paper lacks motivation, it has problems in how the related work is written, and there are many comparisons with key works missing. My decision will therefore be a reject.

**Details Of Ethics Concerns:**

No issues

---

### Official Review · Reviewer_Thjn · 2025-10-30

**Soundness:** 2
**Presentation:** 2
**Contribution:** 2
**Rating:** 2
**Confidence:** 4

**Summary:**

The paper investigates the reliability of HiResCAM, a commonly used deep learning interpretability method that generates attention maps highlighting image regions important for a model’s predictions. The authors theoretically demonstrate  that HiResCAMs are not uniquely determined and admit arbitrary, spurious

To overcome this, the authors propose ContrastiveCAM, a new visualization technique that is invariant to the spurious shift and provides class-versus-class contrastive explanations, offering more precise insight into what differentiates one class from another.

Using these improved explanations, they discover that networks frequently attend to irrelevant image regions. To correct this, they introduce Core-Focused Cross-Entropy, a modified loss function that encourages attention on core (label-relevant) image regions and suppresses attention elsewhere, thereby improving feature alignment between visual regions and class semantics.

Experiments on Hard-ImageNet and Oxford-IIIT Pets show that ContrastiveCAM produces more faithful and robust attention maps, and that Core-Focused Cross-Entropy leads to better predictive performance derived from semantically meaningful regions.

**Strengths:**

The authors identify two main issues in existing methods. From a theoretical perspective, they reveal a limitation of HiResCAM, showing that its attention maps are not uniquely determined. To overcome this, they propose ContrastiveCAM, which eliminates this ambiguity. They also observe that networks often focus on irrelevant image regions, and to address this, they introduce Core-Focused Cross-Entropy, a loss function that encourages attention on label-relevant areas.

Experiments conducted on three different datasets demonstrate the effectiveness of their approach, particularly the Core-Focused Cross-Entropy loss, in improving interpretability and performance.

**Weaknesses:**

The authors discuss the limitation of HiResCAM only from a theoretical perspective, without providing experimental evidence to validate their claim.

Furthermore, they do not convincingly demonstrate that ContrastiveCAM outperforms other modern CAM-based methods. Their comparisons are limited to GradCAM, which is relatively outdated, while many recent and more advanced CAM variants could have been included for a more comprehensive evaluation. In addition, the paper lacks experiments directly comparing ContrastiveCAM and HiResCAM, making it difficult to assess the claimed improvements.

Finally, although the authors claim to have improved feature alignment in convolutional networks, their experiments are conducted only on ResNet. Evaluating their approach on a broader range of convolutional backbones would provide stronger and more generalizable evidence for their conclusions.

**Questions:**

Number of examples used per dataset for quantitative results is unclear — needs clarification.

---

### Official Review · Reviewer_KJVo · 2025-10-30

**Soundness:** 1
**Presentation:** 1
**Contribution:** 1
**Rating:** 0
**Confidence:** 5

**Summary:**

This work introduces an improvement over HiResCAM methodology through contrastive alignement. Experiments are conducted on Hard ImageNet. and Pets and Pascal VOC. It is mostly compated to GradCAM and HiResCAM. ResNet50 backbone is only used for validation.

**Strengths:**

The description of the work is clear. And the attribution-based explanations are important research field.

**Weaknesses:**

This work has a lot of issues that I am worried about:

- first claim is that HiResCAM are popular. Unfortunately I could not find that even this method was published. Only an arxiv version from 2021. And even so, they are cited by 208 works on Google Scholar, and when we will compare it to other CAM methods such as GradCAM (over 30k) and GradCAM++ (over 4k).

- Lack of baselines, GradCAM is a really outdated method, and recent attribution method are LeGrad [1], OMENN [2] or CheferLRP [3].

- Lack of other backbones in experimentations, especially ViTs. Other work do have them.

- Lack of contextualization, broader discussion about LRPs, other CAMs and other attribution-based methods such as B-Cos [4] are missing.

- Visualizations of explanations are not convincing, and there is no experimentation to prove it is better.

- There is no user study to showcase that users better perceive those explanations.

- No XAI benchmarks such as FunnyBirds [5]

[1] Bousselham, Walid, et al. "Legrad: An explainability method for vision transformers via feature formation sensitivity." Proceedings of the IEEE/CVF International Conference on Computer Vision. 2025.

[2] WrĂłbel, Adam, MikoĹ Janusz, and Dawid Rymarczyk. "OMENN: One Matrix to Explain Neural Networks." arXiv preprint arXiv:2412.02399 (2024).

[3] Chefer, Hila, Shir Gur, and Lior Wolf. "Transformer interpretability beyond attention visualization." Proceedings of the IEEE/CVF conference on computer vision and pattern recognition. 2021.

[4] Böhle, Moritz, Mario Fritz, and Bernt Schiele. "B-cos networks: Alignment is all we need for interpretability." Proceedings of the IEEE/CVF Conference on Computer Vision and Pattern Recognition. 2022.

[5] Hesse, Robin, Simone Schaub-Meyer, and Stefan Roth. "Funnybirds: A synthetic vision dataset for a part-based analysis of explainable ai methods." Proceedings of the IEEE/CVF International Conference on Computer Vision. 2023.

**Questions:**

Seeing missing baselines, general not backed up by the literature statements and poor comparisons, I do not got into much methodological details as I do not that see this work can improved during rebuttal period enough to be ready for publishing.

I put more details in the weaknesses section showcasing the limitations of the work, especially contextualization, validation of the method and strong statements.

---

### Meta-Review · Area_Chair_bbQa · 2026-01-07

**Summary:**

All reviewers rejected the submission and there is no rebuttal.

**Reviewer Concerns:**

The main weaknesses reported by reviewers are: the submission lacks strong novelty and empirical support. It focuses on HiResCAM, a non–peer-reviewed and weakly adopted method, without justifying its relevance compared to modern CAM or attribution techniques, and limits comparisons largely to the outdated GradCAM. Experiments are narrow, using only ResNet-based CNNs, with no evaluation on other backbones, ViTs, CLIP, or ImageNet. Explanation quality is not convincingly demonstrated, with unpersuasive visualizations and no quantitative benchmarks, user studies, or XAI evaluations. Theoretical claims are not validated experimentally, direct comparisons between ContrastiveCAM and HiResCAM are missing, and the paper is not contextualized well, failing to clearly differentiate itself from closely related prior work on contrastive or explanation-guided learning.

**Reviewer Scores:**

0,2,2

There is no rebuttal.

---

### Decision · Program_Chairs · 2026-01-26

Reject